# Silicon-Based 3D Microfluidics for Parallelization of Droplet Generation

**DOI:** 10.3390/mi14071289

**Published:** 2023-06-23

**Authors:** Diego Monserrat Lopez, Philipp Rottmann, Martin Fussenegger, Emanuel Lörtscher

**Affiliations:** 1IBM Research Europe—Zurich, Säumerstrasse 4, CH-8803 Rüschlikon, Switzerland; 2Department of Biosystems Science and Engineering, ETH Zürich, Mattenstrasse 26, CH-4058 Basel, Switzerland; philipp.rottman@bsse.ethz.ch (P.R.);; 3Faculty of Science, University of Basel, Mattenstrasse 26, CH-4058 Basel, Switzerland

**Keywords:** three-dimensional microfluidics, droplet generation, silicon, upscaling

## Abstract

Both the diversity and complexity of microfluidic systems have experienced a tremendous progress over the last decades, enabled by new materials, novel device concepts and innovative fabrication routes. In particular the subfield of high-throughput screening, used for biochemical, genetic and pharmacological samples, has extensively emerged from developments in droplet microfluidics. More recently, new 3D device architectures enabled either by stacking layers of PDMS or by direct 3D-printing have gained enormous attention for applications in chemical synthesis or biomedical assays. While the first microfluidic devices were based on silicon and glass structures, those materials have not yet been significantly expanded towards 3D despite their high chemical compatibility, mechanical strength or mass-production potential. In our work, we present a generic fabrication route based on the implementation of vertical vias and a redistribution layer to create glass–silicon–glass 3D microfluidic structures. It is used to build different droplet-generating devices with several flow-focusing junctions in parallel, all fed from a single source. We study the effect of having several of these junctions in parallel by varying the flow conditions of both the continuous and the dispersed phases. We demonstrate that the generic concept enables an upscaling in the production rate by increasing the number of droplet generators per device without sacrificing the monodispersity of the droplets.

## 1. Introduction

Microfluidic systems offer great opportunities for handling, manipulating and characterizing minuscule quantities of liquids, individual particles, droplets or cells by employing channels and other fluid-handling components at micro- or even nanometer scales [1]. Over the past decades, microfluidic chips have been widely applied for diverse applications in the fields of analytical chemistry and biochemistry [2], biological and analysis biomedical analysis [3] and pharmaceutical drug development [4], among others. Many of these applications rely on a large number of objects to be processed per time unit, so achieving high-throughput processes and assays has inherently been a fundamental objective in the microfluidic field [5]. Operating multiple microfluidic devices in parallel may in principle help compensate the lower throughput per single device but usually comes at the costs of a high liquid-routing complexity. Recently, there has been a boost in the development of 3D microfluidic architectures as in those, channels can cross without connecting by going to a so-called redistribution layer which makes them underpassing others [6]. The extension of 2D microfluidic concepts towards 3D enables microfluidic concepts to still perform complex processes while massively increasing the throughput by operating essential components in parallel and still using a single source [7,8,9].

### 1.1. Three-Dimensional Microfluidics

Hitherto, approaches to create 3D microfluidic structures have been proposed mainly—and due to widespread usage—on a conventional polydimethylsiloxane (PDMS) platform, either by layer stacking [10], 3D molding [11] or wax jetting [12]. For example, Headen et al. used soft lithography to create a two-layer, parallel droplet-generator device in PDMS for microencapsulating human mesenchymal stem cells in <100 μm diameter microgels [8]. Despite the low costs and ease-of-fabrication advantages of PDMS, it also presents inherent and fundamental limitations in terms of long-term durability, mechanical compliance and chemical inertness (swelling upon exposure to various solvents or high gas diffusivity). In addition to these PDMS structures, 3D printing has emerged as a novel technique for creating truly three-dimensional architectures at low cost and with an unprecedentedly high flexibility to implement almost arbitrary channel geometries. Most 3D-printed fluidic devices and features up to date, however, are rather on millifluidic-size scales, still demanding for a substantial progress in printing technology, and resist formulations to enter micrometer dimensions [13,14,15]. Nonetheless, some biological analysis applications have already been demonstrated with 3D-printed fluidic devices, such as Li et al., who employed a combination of different materials to simultaneously integrate extraction and concentration components for small-molecule pharmaceuticals from urine, followed by an on-chip electrophoretic separation of the concentrated targets for a quantitative analysis [16]. However, such device had a smallest channel width of 500 μm.

In complementary metal–oxide–semiconductor (CMOS) technology, the 3D stacking of chips has become a habit in the last two decades, enabling the integration density to be increased by going vertical and by using multiple layers for distributing electrical signals and the stacking of memory components. Silicon as a CMOS platform material can be etched almost vertically, and the channels can be closed by a glass cover [17], which renders this approach also attractive for microfluidic applications. Despite the higher processing costs resulting from the manufacturing under clean-room conditions, silicon microfluidics provides an excellent long-term durability, a high mechanical stability and excellent chemical compatibility (chemical inertness towards highly corrosive solvents and low gas diffusivity), conditions that lead to a higher-pressure operation and more uniform flow conditions. The first generations of microfluidic devices correspond to such silicon–glass hybrid structure [18,19] and leverage the highly deterministic CMOS manufacturing processes to define channels and other components down to the a 1 μm feature size. Quite recently, Yadavali et al. benefited from these advantages to develop a silicon-based microfluidic device for the large-scale synthesis of biocompatible microparticles by using a 3D approach and high-pressure operation [9] to overcome the low production rate of microfluidic devices for the generation of microparticles, preventing the translation of many promising microfluidic laboratory-scale results towards commercial-scale production.

### 1.2. Droplet Microfluidics

Droplet microfluidics offers significant advantages for performing high-throughput processes and screening since, unlike in continuous-flow systems, droplet-based microfluidics allows for the independent control of each droplet, thus generating microreactors that can be individually transported, mixed and analyzed [20]. Droplet microfluidic systems can perform a large number of reactions without increasing device size or complexity, and they can even complete simple Boolean logic functions, a critical step towards the realization of a microfluidic computer chip. For this reason, the subfield is sometimes referred to as “digital” microfluidics in order to emphasize the use of discrete and distinct volumes of fluids and to contrast with the continuous nature of other systems [21]. Over the last twenty years, various tools and platforms have been developed in a broad spectrum of fields, leading to applications such as polymerase chain reaction (PCR)-based analyses, enzyme directed evolution, protein crystallization studies and polymeric particle synthesis, among others [22,23,24], all requiring a high throughput to be efficient. More recently, microfluidic droplet generation systems have been used for diverse applications such as fabricating magnetic steerable microrobots [25] or assembling microorganisms within droplet structures for efficient wastewater treatment [26,27].

Accurate and effective droplet generation by using immiscible phases to create discrete soft-matter compartments to encapsulate molecules and cells is critical for all the droplet microfluidic applications. The structures used for droplet generation are either based on a simple T-junction [28] or a triplet junction with three channels joining, referred to as a flow-focusing generator (FFG) [29]. Figure 1a depicts the general design in a top-view with colors describing the compression of the dispersed phase (DP) by the two embracing continuous phases (CPs). There exist three main breakup modes or mechanisms that lead to droplet formation [30,31]. The squeezing mechanism consists on the DP blocking the flow of the CP leading to a rise pressure of the latter, which squeezes the DP to the point of breaking it off into single droplets; this mechanism is also referred as “geometry-controlled” in the literature [32,33]. The dripping mechanism, however, is due to the shear stresses overcoming the interfacial tension; therefore, the CP shears the DP into droplets [34]. Finally, droplets can also be formed by jetting due to the Plateau–Rayleigh instability, which can happen when the DP forms long threads within the CP [35]. Additionally, there also exist two more passive droplet generation mechanisms studied in the literature that are tip-streaming [36] and tip-multibreaking [37]. The operation under the different modes mainly depends on the value of the dispersed and/or continuous phase capillary numbers, which is the dimensionless quantity defined as the ratio of viscous to interfacial forces [31]. In addition to microfluidic-based droplet generation, there also exist other methods such as phase separation or emulsion curing to create droplet-embedded structures in soft materials [38].

The microfluidic channels to feed an FFG come from two inlets, one for CP and DP each, as schematically shown in Figure 1b. When upscaling to an array of FFGs with the goal of increasing the system’s droplet generation throughput, it becomes obvious that multiple inlets must be used not to cross channels. This is the case for either case of sourcing, with the CP or the DP as a single source (Figure 1c,d). Instead, we propose a redistribution layer to be used, to feed a single FFG via a vertical interconnection channel or via (Figure 1e), which resolves the spatial constraints and enables a scaling of FFGs to large arrays by using the redistribution layer to let channels underpass others (Figure 1f). In this paper, we present a fabrication route to create 3D silicon–glass microfluidic devices that describe upscaling strategies to operate multiple droplet-generation devices in parallel using a single input source. The 3D architecture realized in silicon resembles the layer stacking of PDMS, where the microfluidic channels are patterned at different layers and vertically interconnected. Similarly, one of these additional layers is used as a redistribution layer to enable channels to cross without connecting. To demonstrate the potential of this architecture, several devices with one up to eight flow-focusing droplet generators connected in parallel are fabricated and characterized comprehensively in terms of droplet size and uniformity. The concept is highly generic and can be used to operate multiple microfluidic components other than droplet generators in parallel to increase the throughput per chip. In the following, we describe how these devices can be fabricated and characterize their operation in droplet-generating tasks.

## 2. Results and Discussion

### 2.1. Microfabrication Route for Silicon 3D Microfluidics

To realize the 3D microfluidic network with multiple FFG devices depicted in Figure 1f, we fabricated a three-layer structure composed of a glass–silicon–glass sandwich. All the microfluidic channels and structures were patterned into the middle silicon wafer, as detailed in the fabrication sequence in Figure 2a. The two bottom and top glass wafers behaved as sealing structures for the Si channels. While the top glass wafer was unpatterned, the one on the bottom was equipped with through holes (500 μm in diameter) machined to provide mass-flow access via the microfluidic inlets and outlets. Within the silicon middle wafer, there were three different layers, labeled according to their functionalities: operation layer, redistribution layer and interconnection layer. These layers are schematically represented in Figure 2b. The operation layer was etched on the top side of the wafer, the interface where all the processes of interest take place and are visualized. For the devices presented in this work, it was the layer where multiple FFGs were located for the droplet generation. In contrast, the redistribution layer was etched on the bottom side of the silicon wafer and comprised different channels in parallel. This layer served the purpose of distributing a reagent to the different points of interest to access the operation layer. Finally, the interconnection layer was created by etching vias through the whole thickness of the silicon wafer, at desired points. This layer had the function of connecting the operation and the redistribution layer. Figure 2c shows in detail a 50 μm deep etched operation layer on the silicon wafer, where the vias that connect to the redistribution layer can also be observed. The insert shows in detail how the flow-focusing junction looks, whose exact geometric parameters can be found in Section 4. All the devices were fabricated using the same parameters for the junctions’ geometry, as the main goal of this paper was to study the effect of several junctions in parallel and not the impact of geometric modifications on the junctions, as it has been thoroughly reported in literature [39,40].

While the introduction of a redistribution layer resolves the channel-crossing issues, it provides an additional level of complexity for the microfluidic layout, as the vias need to be taken into account for the fluid-dynamic conditions and backpressure regulation across the 3D network. In particular, if the goal is to generate the most uniform droplets based on multiple FFGs operated in parallel and being fed from a single DP and CP source each, it is key to have a uniform pressure distribution in all three channels leading to the FFGs such that equal flow rates are present at each junction. These conditions can be achieved either by creating fully symmetric flow distributions by a rational design or by using flow resistors prior to the FFG device [9]. While the high-aspect flow resistors provide an excellent mean to locally regulate the pressure by the constraint introduced, the concept reduces in return the maximum flow rate possible and leads to quite high pressure build-ups across the entire device. For these reasons, we developed a fully symmetric flow-distribution layout by implementing a fractal ramification of channels across operation and redistribution layers. In addition to being symmetric, the width of each channel was also divided equally every time the channel split into two new ones, in a way that the fluidic flow was kept constant through the microfluidic network towards each FFG, as depicted in Figure 1f. After the FFG, the channels widened up until they merged prior to exiting the device through the single existing outlet. From the FF1 layout which only contained one FFG but already used the 3D structure for the dispersed phase trough the redistribution layer until the junction, up to two, four and eight FFGs were implemented according to the design rules described above, leading to the FF2, FF4 and FF8 layouts. For the experimentation with those devices, we built a setup consisting of a tailored 3D-printed microfluidic interface to easily access the device’s inlets and outlet. This interface could be connected to pump-actuated syringes via PEEK tubing and placed under a microscope which was equipped with a high-speed camera for visualization of the droplet generation processes. This setup is shown in Figure 3 and described in more detail in Section 4.

### 2.2. Droplet Size and Production Rate Dependence on Relative CP and DP Flow Rates

We first characterized the operation of a single FFG device, that was used later on as a reference when describing the properties of multiple FFGs operated in parallel. While keeping the CP flow rate, $\phi_{c}$, constant at 1500 μL/h, we continuously varied the DP’s flow rate, $\phi_{d}$, from 500 μL/h to 5000 μL/h. We defined the relative flow rate ratio between both phases as $\phi_{d}$:$\phi_{c}$, and the range from 1:3 to 10:3 was explored. Figure 4a shows images of the droplet production for the different conditions tested. The quantitative results for the droplet size and droplet production rate are depicted in Figure 4b,c, as well as in Table 1. The droplet diameter increased linearly as the ratio $\phi_{d}$:$\phi_{c}$ increased. This result obeyed the well-known scaling law proposed by Garstecki et al. for T-junctions [41], which describes the droplet size dependence on the ratio between the CP and DP flow rates when working in the squeezing droplet-generation regime (at low capillary numbers). The squeezing mechanism also explains why the droplet sizes are bigger than the width of the junction where they are created [33]. As explained in Section 4, the viscosity and density values of the oil used in these experiments (manufacturer: Bio-Rad, Hercules, CA, USA) were proprietary, which prevented us from calculating the exact capillary number for these and the upcoming experiments. Instead, the droplet production rate did not increase in a proportional linear way with the linear increase of the disperse phase flow but rather had a logarithmic growth (see Figure 4c). This was a consequence of the bigger droplet size for higher flows, which implied a larger amount of dispersed material in each droplet. Finally, a set of parameters requires a special discussion as the results fell out of the expected behavior. When increasing the DP to 5’000 μL/h (which corresponds to a ratio of 10:3), the total flow used for a single junction created a built-up backpressure that affected the droplet generation process leading to droplet sizes bigger than expected. For our design, we defined big droplets as droplets with a diameter larger than 100 μm (which is twice the width of both the DP and CP channels).

Next, we performed the same study using the FF2 device, which had two FFGs, to evaluate how the relative ratios between CP and DP affected the production of droplets when having parallel junctions sharing the same inlets and outlet. Figure 5a shows the images of the different experiments performed. In that case, we evaluated three different values for the ratio $\phi_{d}$:$\phi_{c}$ (1:3, 1:1 and 2:1), and we used three different flows for the continuous phase, which we referred to as low, medium and high rates, corresponding to 1’500, 3’000 and 4’500 μL/h, yielding nine different sets of parameters in total. The medium rate of 3’000 μL/h was equivalent to the flow rate used in the previous experiments with the FF1 device, as that flow was now split among the two parallel FFGs of the device (1’500 μL/h per junction). The middle row of Figure 5a corresponds to the experiments with that medium flow rate, and the results were similar to those obtained with the FF1 in value and trend (increasing droplet size for an increased $\phi_{d}$:$\phi_{c}$ ratio), therefore following the Garstecki scaling law previously introduced. Meanwhile, when working at low CP flow rates (top row of Figure 5a), it was not possible to achieve droplets of small size (with a diameter below 100 μm) despite increasing the DP flow. The droplet production directly transitioned from very irregular for the low $\phi_{d}$:$\phi_{c}$ ratio to big droplets for higher ratios. Nooranidoost et al. studied the effect of the flow rate ratio on the formation of Newtonian droplets in a viscoelastic ambient fluid and predicted how using flow rate ratios with a CP much higher than the DP (three times bigger or more) could lead to the generation of highly irregular droplets [42], which was exactly the case observed here. Finally, when working at high CP flows (bottom row of Figure 5a), a high throughput of homogeneous droplets could be achieved for a flow ratio of 1:1 or lower with similar behaviors (and sizes) similar to the mediums flows but at an obvious increased production rate. However, when increasing the DP flow to a 1:2 ratio, the system was not stable anymore, and the produced droplets had an inhomogeneous size. Similarly to what we observed with he FF1 device, the high total flow reached in that last scenario came together with a high built-up backpressure that affected the stability of the droplet generation leading, in that case, to an unequal process on the two parallel junctions.

Figure 5b,c show the droplet size’s quantitative results for the different sets of parameters studied. From those graphs, it can be concluded that, when working with FFGs in parallel, having either too low flow rates or too high flow rates might make the system not behave stably depending on the ratio between CP and DP flows. However, in the medium regime, the system worked more stably and it was even possible to control the droplet size by controlling the relative ratio of the flow rates.

### 2.3. Maximum Flows for Parallel Junctions’ Operation

Normal device operation should be in a flow velocity regime where all the parallel junctions produce uniform droplets. From the previous study with the FF2 device, it is understood that besides the relative ratio between the flow rates, the total flow employed also has a high impact on the droplet homogeneity. While in the squeezing mechanism, the interfacial forces dominate the shear stresses, meaning that the break-up dynamics is dominated by the pressure drop across the droplet as it forms, the use of high flow rates might force a transition to a jetting regime where no droplets are formed and the inner fluid simply jets through the continuous phase. It is important to note that this jetting concept is different form the jetting regime where droplets are produced by the Plateau–Rayleigh instability as introduced in Section 1.2. Droplet generation by jetting requires a CP flow rate much higher than the DP and has been studied for $\phi_{d}$:$\phi_{c}$ ratios from 1:10 to 1:400 [29,35]. Thus, we studied with all our sets of devices (FF1, FF2, FF4 and FF8) how much the flow rates could be increased before the transition from regular droplet generation by squeezing to a jetting scenario where no droplets are formed occurs. We defined the total flow as $\phi_{t}=\phi_{d}+\phi_{c}$ and chose the ratio $\phi_{d}$:$\phi_{c}$ equal to 1:1 for that set of experiments. For each one of the experiments, the total flow was progressively increased until the jetting was achieved in at least one of the junctions; nevertheless, other anomalous behaviors were observed and noted. Just before jetting was reached, there was a regime where droplets were formed in a very irregular way and the droplet generation was not stable anymore. Furthermore, it was observed that droplet size also depended on the total flow as bigger droplets were formed for a higher total flow. We also registered when the transition to big droplets (diameter larger than 100 μm) occurred for each device. In that case, we registered the transition to big droplets when they had a diameter larger than 100 μm. The quantitative results are summarized in Table 2 as well as graphically shown in Figure 6b. Moreover, Figure 6c shows images of the junction working at each one of the behaviors described.

These results show that the transition from droplet generation to jetting (green to blue area in Figure 6b) did not follow a linear trend. Despite the fact that an increase in the number of parallel junctions increased the maximum total flow value before reaching jetting conditions, this increase was not directly proportional to the number of parallel junctions. The channel widening introduced right after the FFG junction was aimed at reducing the built-up backpressure; however, the final outlet dimensions were identical for all devices. Thus, the FF8 device could not stand a flow eight times larger than the maximum flow allowed by the FF1 device, as this built-up backpressure affected the droplet generation process and eventually destabilized one or more of the eight parallel junctions when using high flows.

### 2.4. Droplet Generation Homogeneity

In addition to exploring the flow limits for parallel operation of FFGs, we were also interested in exploring how the scaling up worked when operating at a moderate flow velocity regime. Therefore, starting with flow rates of $\phi_{c}$ = 1000 μL/h and $\phi_{d}$ = 1000 μL/h for the FF1 device, we produced and collected droplets for each device using proportionally increasing flows for each device (see Table 3). Subsequently, we placed the droplets on a microscope’s counting slide to take images for further analysis as detailed in Section 4. The results are shown in Figure 7a,b. The average droplet size obtained with each device was basically the same (around 65 μm), while the width of the distribution fitting the histogram did not increase significantly when increasing the number of parallel junctions (only from FF1 to FF2, there was a considerable increment as could be expected). These results proved that, when operating at medium velocity regimes, the droplet generation process could be scaled upon maintaining a high droplet quality and homogeneity. According to the literature, bulk storage and transfer of droplets might affect the monodispersity of a droplet population in the long term; therefore, these results could be further improved by on-chip storage in fluidic microcavities [43].

Finally, we wanted to study if scaling up while maintaining a reasonable homogeneity of the droplets was also working for obtaining a high throughput of smaller droplets. In Section 2.2, we learned how using flow ratios with a higher CP flow than DP flow led to the production of smaller droplets. Therefore, we used a flow ratio of $\phi_{d}$:$\phi_{c}$ equal to 1:4 with the FF1 and FF8 devices. Figure 7d shows the histogram and fitted distribution obtained as well as a micrograph of the droplets collected, while the numerical results are also included in Table 3. Using such parameters, it was possible to obtain highly homogeneous droplets with an average size around 46 μm. Given that the dimensions of the flow-focusing junction’s orifice width was 40 μm ($W_{o}$), we assumed that the droplet generation mechanism was still in the squeezing regime under those parameters. Increasing the CP flow rate even more might lead to even smaller droplets and potentially work in the dripping regime, but it would require a considerable higher amount of CP material without increasing the total droplet throughput substantially. Therefore, whenever the aim is to produce small droplets, the most critical and effective parameters to adjust are the dimensions of the flow-focusing junction as it has been proven to be the most influential parameters [31,40].

## 3. Conclusions and Outlook

To summarize, a microfabrication route was proposed to achieve glass–silicon–glass 3D microfluidic structures based on the implementation of vertical vias and a redistribution layer. Its operation was demonstrated by the fabrication of droplet generation devices with several flow focusing junctions in parallel. Those junctions were supplied by using only a single source for the dispersed phase and one source for the continuous phase. The channel crossing unavoidable in 2D was avoided thanks to the vias and the redistribution layer. Through the rational design and by experimental optimization, a balanced flow distribution among the different parallel channels was achieved, enabling a stable and reproducible droplet production on each junction without the use of flow resistors. It was further determined experimentally how different flow ratios between the two phases impacted the droplet generation production process and how it compared to one junction that could be realized without going to a 3D design. Furthermore, the system was tested towards the maximum flow rates feasible given its backpressure limitations, leading to a decrease in the droplet uniformity at the highest flow rates. At medium flow rates, the results were found to be highly reproducible and the system’s operation could be configured, as its followed a simple scaling behavior defined by the number of parallel generators used.

The 3D extension of silicon microfluidics presented here was combined with the high accuracy and reproducibility of silicon enabled by the underlying manufacturing technologies, making the concept attractive towards industrial use. Similar to their electronic counterparts developed over the last four decades in CMOS technology, silicon microfluidic devices can, in principle, be equally upscaled in 2D to large dimensions and in 3D—as shown above—to multiple layers. This provides many degrees of freedom for microfluidic platform design on a deterministic and solid-state platform that is fully mass-production-compatible. In addition, silicon-based microfluidic devices show an excellent long-term durability and a high mechanical stability, conditions that can handle higher pressure operations while still yielding uniform flow conditions and an improved operation reproducibility. These properties become highly relevant when working with high-viscosity fluids [44,45]. The high chemical compatibility of silicon towards corrosive solvents can be leveraged for multiple applications requiring nonaqueous solutions for operation. For instance, silicon microfluidic silicon devices have been used for the high-precision production of polymeric giant unilamellar vesicles to encapsulate enzymes [46]. This process benefits from the solvent compatibility of silicon to avoid channel swelling, as well as from the multiple ways to control its surface chemistry. Further, it makes devices reusable as they can be cleaned under harsh conditions. Moreover, silicon technology enables a scalable, high-density integration of other functionalities such as channel-nearby electrodes that are not feasible using other technologies [47], sensing components, optoelectronic components, including laser sources and photodetectors, that can all be seamlessly and monolithically integrated in silicon. This setting will pave the road towards numerous novel applications in fundamental research, chemistry, healthcare and life sciences or pharmaceutical production.

## 4. Material and Methods

### 4.1. Sample Fabrication

Double-side polished silicon wafers with a 100 mm diameter and 300 μm thickness were purchased from University Wafer, while Borofloat33 glass wafers with a 100 mm diameter and 700 μm thickness were obtained from Wafer Universe. The different chrome masks with each one of the designs for the photolithography steps were produced using a Heidelberg DWL2000 laser writer. Both sides of the silicon wafer were patterned via UV photolithography (using Microchemicals AZ4533 photoresist and a Süss Mask Aligner M6 tool) and the deep reactive etching (DRIE) was performed with an Omega Rapier DRIE module from SPTS. The channels and structures patterned on each side had a depth of 50 μm. A third photolithographic step was performed (this time with a thicker AZ4562 photoresist) followed by a longer etching process to yield at least 200 μm in depth, to interconnect structures from both sides of the silicon wafer. The holes on one of the glass wafers were drilled using a 0.5 mm diameter diamond grinding pen (Model 1A1W from Haefeli) mounted to a CNC tool. After proper cleaning with a piranha solution, this wafer was bonded to the bottom side of the silicon wafer through an anodic bonding process at 480 ${}^{{}^{\circ}}$C and 1.2 kV performed by a homemade wafer-bonding tool as described in previous work [48]. Subsequently, the same tool was used to bond an unpatterned glass wafer to the top side of the silicon wafer. The glass–silicon–glass sandwich was then diced into individual devices using an ADT ProVectus LA 7100. A total of 20 devices with a size of 15 mm × 15 mm each were obtained per wafer stack. The design of the wafer included 5 copies of each kind of device according to the number of parallel droplet generation junctions (FF1, FF2, FF4 and FF8). The geometric parameters of the flow-focusing junction, as defined in Figure 1a, were: $W_{DP}$ = 50 μm, $W_{CP}$ = 50 μm and $W_{o}$ = 40 μm, while the value for the height of the channel was *H* = 50 μm.

### 4.2. Microfluidic Setup

The microfluidic interface was designed with CATIA V5R20 and 3D-printed by stereolithography using a Formlabs Form 3. The interface microfluidic ports were compatible with M6 microfluidic fittings and were connected to syringes via PEEK tubing with an inner diameter of 0.5 mm. Markez FFKM O-rings were used to seal the contact between the devices’ surface and the tubing ferrules. For the experimentation, the interface was mounted on an optical stereo-microscope, and the syringes were actuated by neMESYS (Cetoni GmbH) syringe pumps. A high-speed camera (MotionBLITZ EoSens mini 1) connected to the microscope was used to capture videos of the device operation. QX200 droplet generation oil from Bio-Rad was used as the continuous phase for the droplet generation experiments, while DI-water was used as the dispersed phase. The values of the viscosity and density of the oil were proprietary information from Bio-Rad, which prevented us from calculating the exact values for the capillary number. The silicon microfluidic devices were under oxygen plasma for 3 min at 600 W and treated with Sigmacote (a siliconizing reagent for glass and other surfaces) for 5 min before being used for the first time.

### 4.3. Image and Video Droplet Analysis

For the droplet-size homogeneity experiments (Figure 7), around 1 mL of water in oil droplets was collected in a vial during the experimentation with each device. A few microliters of droplets were pipetted into a cell-counting slide in order to take bright-field microscope images using a Leica VZ700 C microscope. This process was repeated several times for each collection in order to have a representative subset of the population. Five or six images containing around 100 droplets each (for droplets with a diameter of 65 μm) or around 200 droplets (for diameters of 45 μm) were analyzed (having a minimum number of 600 droplets per experiment) using Fiji ImageJ software to determine the diameter of all droplets by image recognition. The circle Hough transform (CHT) algorithm was used as a well-established extraction technique in image processing for detecting circles [49]. Previously, each image was preprocessed by converting it to a 32-bit grayscale image in order to apply an intensity threshold to segment the image to a binary image that could be used for the CHT. The results obtained (diameter for each circle found in the image) were plotted as histograms and fitted to a Gaussian distribution to obtain the mean and FWHM values.

For the relative flows experiments (Figure 4 and Figure 5), the analysis was performed directly on images extracted from the videos recorded during operation. Videos were recorded at 3000 fps and exported at 30 fps for each combination of flow parameters tested. Therefore, one second of those videos corresponded to 10 ms of real-time device operation, and single-droplet generation processes could be clearly observed (individual frame images were accessible one by one). To determine the droplets’ size, different frames for each experiment were analyzed using the Measure tool within Fiji ImageJ software. The channel width exciting the FFG junction was taken as an internal scale bar (100 μm). To determine the droplet production rate, droplets generated during 10 s of video (100 ms of real time) were manually counted and the value obtained was converted to droplets/s in real time.

## Figures and Tables

**Figure 1 micromachines-14-01289-f001:**
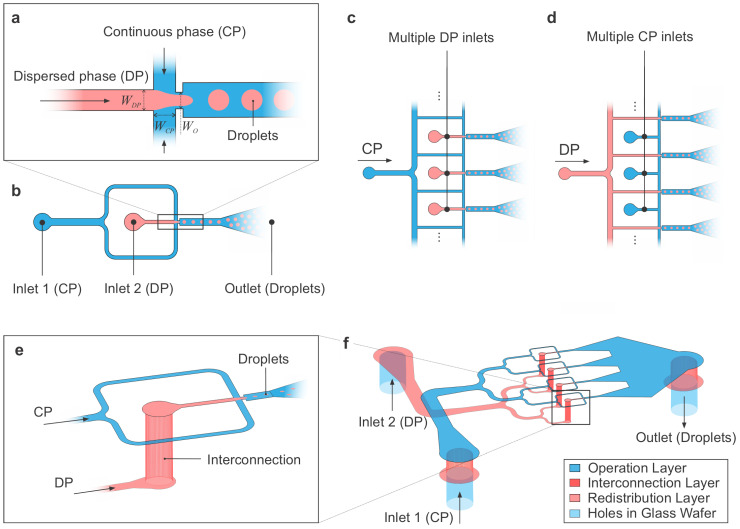
**Concept of flow-focusing-based droplet generators in 2D and 3D arrangements:** (**a**) Schematic representation (top-view) of a flow-focusing-based (droplet) generator (FFG) with one dispersed phase (DP) and two continuous phases (CP) merging to produce droplets. The generic parameters that define the geometry of the junction are the DP inlet width ($W_{DP}$), CP inlet width ($W_{CP}$), orifice width ($W_{o}$) and overall channel height (*H*). (**b**) Typical microfluidic layout with two inlets to source the FFG and one outlet used for droplet collection. (**c**) Two-dimensional arrangement of multiple FFGs with the CP sourced from a single inlet while the DP is sourced from multiple inlets, realized via external splitting. (**d**) Two-dimensional arrangement of multiple FFGs with the DP sourced from a single source while the CP is sourced from multiple inlets. To achieve a single-source configuration, the in-plane spatial constraints would result in channel crossings. (**e**) Proposed 3D structure with interconnects and DP being fed (and distributed) from an underpassing layer (3D view). (**f**) Full scalability of multiple FFGs operated in parallel in the 3D embodiment, shown exemplarily for the case of four FFGs with a symmetric, branchial channel layout on both layers to ensure a symmetric flow distribution and pressure homogeneity on each path towards the FFGs. The 3D extension allows both CP and DP to be fed from a single source.

**Figure 2 micromachines-14-01289-f002:**
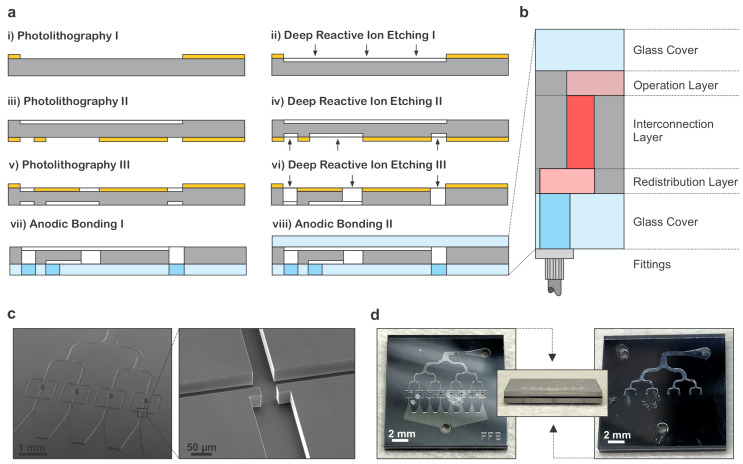
**Generic fabrication route and processing results:** (**a**) Microfabrication processing steps to achieve glass–silicon–glass 3D microfluidic devices, realized by DRIE from two sides to create both channels and interconnects across the Si layer. Eventually, the Si chip is anodically bonded to a glass cover comprising openings as inlet and outlet ports. (**b**) Cross-sectional view of the entire chip stack with color coding as in Figure 1f. The design deploys the bottom Si interface as a redistribution layer such that the top operation layer remains optically accessible (see Figure 3). (**c**) SEM images of the operation layer with a layout that comprises four FFG (FF4) devices (left) with details of the FFG design and almost vertical side walls (right). The images were all acquired prior to bonding for a channel depth of 50 μm. (**d**) Optical micrographs of a final chip with a size of 15.0 mm × 15.0 mm × 1.9 mm comprising a layout with eight FFG (FF8) devices (left), where the glass–silicon–glass wafer stack can be observed in a cross-sectional view (middle) with the bottom interface (right) depicting the inlets and outlets for fluid access.

**Figure 3 micromachines-14-01289-f003:**
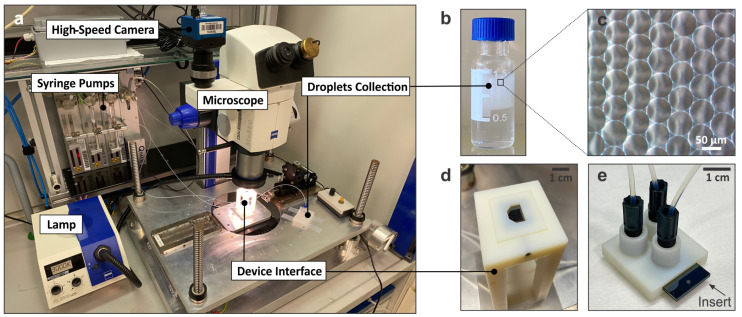
**Microfluidic setup:** (**a**) Optical image depicting all the components required for conducting the experiments presented in the text, including syringe-based dosing for both DP and CP, a high-speed camera and a 480× stereo-microscope with top illumination. (**b**) A 1.5 mL vial used to collect droplets, with the latter accumulating after a few minutes on the top such that the emulsion can be seen by the naked eye. (**c**) Optical micrograph showing densely packed droplets with uniform sizes. (**d**) Three-dimensional-printed device interface and holder allowing optical access from the top with high-NA objectives. (**e**) Device interface where the chip is inserted through a slit and fixed and sealed by tightening the fittings with O-rings (not visible).

**Figure 4 micromachines-14-01289-f004:**
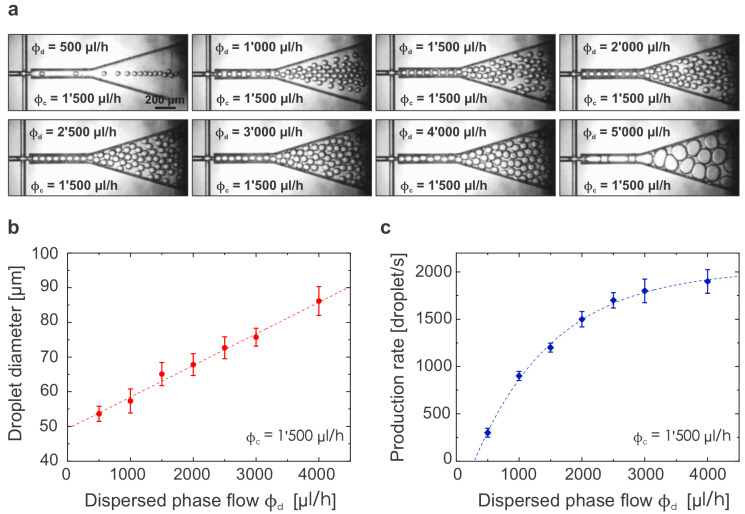
**Characterization of the FF1 device with a varying $\phi_{c}$:$\phi_{d}$ flow ratio:** (**a**) Screenshots extracted from high-speed videos acquired upon operating an FF1 device under the flow conditions labeled. The CP flow was kept constant at 1’500 μL/h for all experiments ($\phi_{c}$ = 1’500 μL/h). (**b**) Droplet diameter (mean ± standard deviation) as a function of DP flow rate. The increase in droplet diameter follows a linear behavior up to the DP rate of 4’000 μL/h, with the fitted function: y=49.31+0.0091x. (**c**) Droplet production rate (defined as the number of droplets generated per second) as a function of DP flow rate. The rate asymptotically approaches a maximal threshold of approx. 2’000 droplets/s for the FFG design used and under the conditions described. The fitted function was $y=2031.3-2538.5\times e^{-x/1287.8}$.

**Figure 5 micromachines-14-01289-f005:**
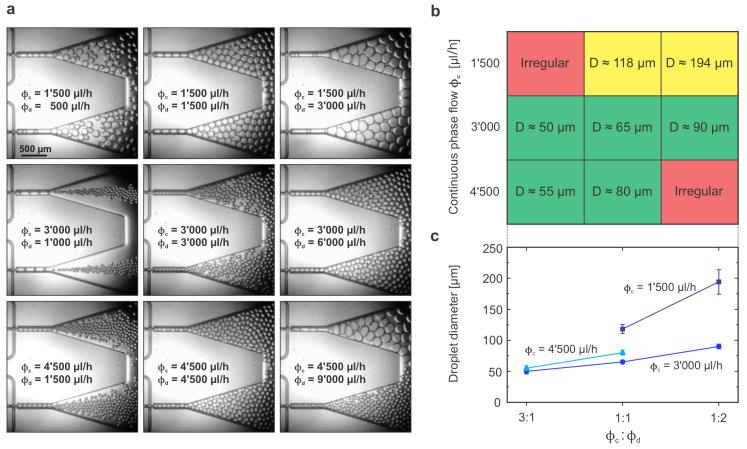
**Characterization of the FF2 device with a varying $\phi_{c}$:$\phi_{d}$ flow ratio:** (**a**) Screenshots extracted from high-speed videos acquired upon operating an FF2 device at the parameters given. Both outlets could be simultaneously imaged. For each row, $\phi_{c}$ was kept constant at 1’500, 3’000 or 4’500 μL/h (from top to bottom), while $\phi_{d}$ was adjusted proportionally in each column to achieve the following ratios: $\phi_{c}$:$\phi_{d}$ = 3:1, 1:1 and 1:2 (from left to right column). (**b**) Droplet production characteristics as qualitative (irregular regime; red) and quantitative (droplet diameter for uniform regimes; green for small droplets; yellow for large droplets) map. (**c**) Plot of measured droplet diameters for the various flow parameters explored.

**Figure 6 micromachines-14-01289-f006:**
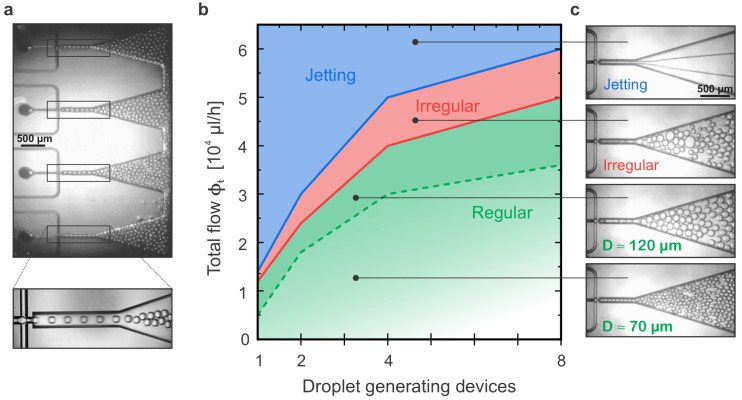
**Characterization of multiple FFGs for maximum flows:** (**a**) Simultaneous tracking of droplet production in the FF8 device with an objective capable to image a region of 3 mm × 5 mm (top). Detailed zoomed-in image of droplet production at a single junction. (**b**) For each device, CP and DP flow rates were increased while keeping a $\phi_{c}$:$\phi_{d}$ flow ratio of 1:1 in order to determine the transition from regular operation (green area) to jetting regime (blue area) where no droplets are produced. Before the jetting occurred, the transition to irregular droplet size production was determined (red area) as well as the regime where large but uniform droplets could still be generated (diameter above 100 μm). Plotted is the total flow defined as $\phi_{t}=\phi_{c}+\phi_{d}$. (**c**) Single-device video snapshots to illustrate each one of the regimes described.

**Figure 7 micromachines-14-01289-f007:**
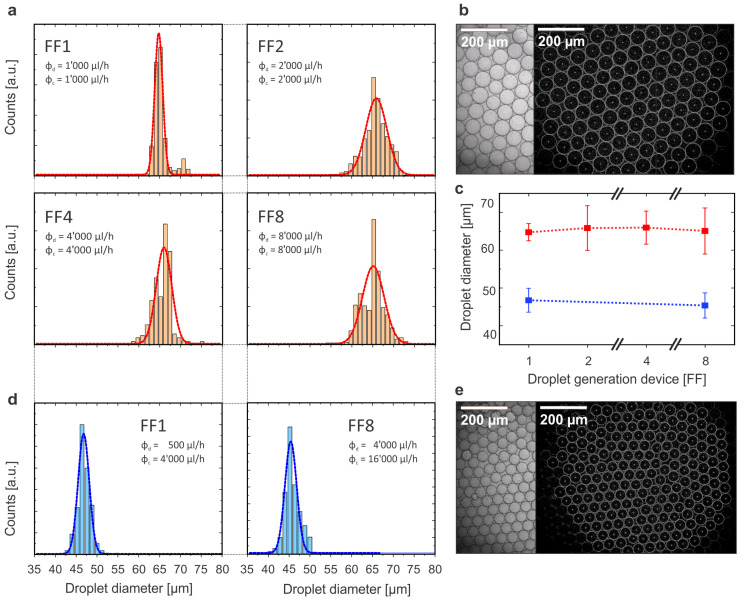
**Production results and statistics across all FFGs operated:** (**a**) Droplet-size distributions with Gaussian fits of the histograms for the experiments conducted with parameters described in Table 3 to yield droplets around 65 μm in diameter. (**b**) Optical bright-field micrograph of the collected droplets (left), with the same image after image recognition analysis was performed (as described in Section 4) and the identified circles are overlaid over the processed image (right, dark image). (**c**) Droplet mean diameter and full width at half-maximum as error bar for an increasing number of FFGs operated in parallel. Red dots and line correspond to the results from subfigure a, while blue dots and line correspond to the results from subfigure d. (**d**) Droplet-size distributions with Gaussian fits of the histograms for the experiments conducted with parameters described in Table 3 to yield droplets around 46 μm in diameter. (**e**) Optical micrographs of these smaller droplets before and after image recognition measurements were performed.

**Table 1 micromachines-14-01289-t001:** Flow rates used for the experiments with FF1 shown in Figure 4 to study the impact of relative flows between CP and DP on the droplet production rate and the droplet size.

CP Flow ($\phi_{c}$)	DP Flow ($\phi_{d}$)	Droplet Diameter	Production Rate
(μL/h)	(μL/h)	Mean ± SD (μm)	(droplets/s)
1’500	500	53.6 ± 2.2	300 ± 47.14
1’500	1’000	57.3 ± 3.5	900 ± 47.14
1’500	1’500	65.1 ± 3.3	1200 ± 47.14
1’500	2’000	67.8 ± 3.2	1500 ± 81.65
1’500	2’500	72.7 ± 3.2	1700 ± 81.65
1’500	3’000	75.7 ± 2.6	1800 ± 124.72
1’500	4’000	86.2 ± 4.2	1900 ± 124.72
1’500	5’000	188.4 ± 11.8	600 ± 47.14

**Table 2 micromachines-14-01289-t002:** Total flows ($\phi_{t}=\phi_{d}+\phi_{c}$) with the ratio $\phi_{d}$:$\phi_{c}$ = 1:1 at which the transition from normal operation to big droplets (diameter > 100 μm), then irregular droplets and finally jetting was observed.

	Total Flows (μL/h) for Transition to:		
**Device**	**Big Droplets**	**Irregular Droplets**	**Jetting**
FF1	5’000	12’000	14’000
FF2	18’000	24’000	30’000
FF4	30’000	40’000	50’000
FF8	36’000	50’000	60’000

**Table 3 micromachines-14-01289-t003:** Flow rates used and droplet size results for the droplet homogeneity experiments with the different devices. The corresponding normal distributions are shown in Figure 7a.

Device	CP Flow ($\phi_{c}$)	DP Flow ($\phi_{d}$)	Droplet Diameter
	(μL/h)	(μL/h)	Mean ± FWHM (μm)
FF1	1’000	1’000	64.79 ± 2.26
FF2	2’000	2’000	65.90 ± 5.91
FF4	4’000	4’000	66.02 ± 4.38
FF8	8’000	8’000	65.12 ± 6.09
FF1	4’000	500	46.79 ± 3.14
FF8	16’000	4’000	45.40 ± 3.32

## Data Availability

The data presented in this study are available on request from the corresponding author.

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
