# Peer review of "Silicon-Based 3D Microfluidics for Parallelization of Droplet Generation"

_micromachines, 2023, doi:10.3390/mi14071289_

Round 1

Author Response

Please see the PDF attachment.

Reviewer 2 Report

Review of the manuscript entitled “Silicon-based 3D Microfluidics for Parallelization of Droplet Generation”. The authors have done practical work about developing a generic fabrication route based on the implementation of vertical vias and a redistribution layer to create glass-silicon-glass 3D microfluidic structures with multiple parallel flow-focusing junctions. And the system construction, droplet generation, the relationship between droplet size and dispersion and continuous phase flow rate, and the uniformity of generated droplets were verified in detail. We would strongly recommend that this work should be published in Micromachines. To further improve the quality of this work, we hope the authors will consider the following suggestions.

1. From lines 26 to 33, there seems to be a lack of citations to enrich the background knowledge of the development of 3D microfluidics to promote the development of parallel operation of multiple microfluidic devices.

2. Some previous studies have proposed the droplet generation device with several parallel flow nodes supplied by a source for high-throughput droplet generation, and proved the feasibility of the device through practical application. Please explain the advantages of the method proposed in this paper. And there are two recommended references (Liu, Y., et al., Benefited wastewater utilization via configurable, spatialized, and microorganisms-integrated biophotonic systems. CEJ (2023); Chen, L., et al., Reconfigurable modular microbiota systems for efficient and sustainable water treatment. CEJ (2022)).

3. In line 220, the authors said “It is not possible to achieve droplets of small size despite increasing the DP flow”, and how to define “small size”?

4. In the Results and Discussion section, the authors made a detailed study and discussion on the processing of the 3D microfluidic FFG device, the relationship between the size and yield of the generated droplets and the flow rate of CP and DP, the maximum flow rate of the device operation, and the uniformity of the generated droplets. It will be better if the authors could increase the research and discussion of some applications that can be carried out using the device.

5. In Figure 3b, the droplets were collected in a 1.5 ml vial, and it is said that the dispersed phase was DI-water and the continuous phase was oil in Section 4.2, please clarify how the resulting droplets are collected into the 1.5ml vial and maintain their shape and size.

6. There are some minor formatting issues in the manuscript. For example, the repetitive “the” in line 101; the punctuation symbol problem of “Φd : Φc, 1:3, 1:1, and 2:1” in line 211, etc. Please revise it.

Author Response

Please see the PDF attachment.

Reviewer 3 Report

Nil

Author Response

Please see the PDF attachment.
